# Strong Association Between MiRNA Gene Variants and Type 2 Diabetes Mellitus in a Caucasian Population

**DOI:** 10.3390/ijms262110447

**Published:** 2025-10-28

**Authors:** Eleni Manthou, Xanthippi Tsekmekidou, Fotis Tsetsos, Theocharis Koufakis, Maria Grammatiki, Pantelitsa Rakintzi, Eirini Melidou, Georgios Karaliolios, Peristera Paschou, Nikolaos Papanas, Kalliopi Kotsa

**Affiliations:** 1Diabetes Center, First Department of Internal Medicine, Aristotle University of Thessaloniki, AHEPA University Hospital, 54636 Thessaloniki, Greece; elenamanthoukk@gmail.com (E.M.); xanthippitsekmekidou@gmail.com (X.T.); grammatikimaria@gmail.com (M.G.); rakitzi.endo@gmail.com (P.R.); melidou.eirini@gmail.com (E.M.); gdocthes@gmail.com (G.K.); 2Department of Molecular Biology and Genetics, Democritus University of Thrace, 68100 Alexandroupoli, Greece; fotis.tsetsos@gmail.com (F.T.); ppaschou@purdue.edu (P.P.); 3Second Propaedeutic Department of Internal Medicine, Hippokration General Hospital, Aristotle University of Thessaloniki, 54642 Thessaloniki, Greece; thkoyfak@hotmail.com; 4Diabetes Center, Second Department of Internal Medicine, Democritus University of Thrace, 68100 Alexandroupoli, Greece; papanasn@med.duth.gr

**Keywords:** type 2 diabetes, SNPs, miRNAs, genetic predisposition

## Abstract

MicroRNAs (miRNAs), small non-coding RNAs, have emerged as promising diagnostic and prognostic biomarkers for various diseases. However, their role in the pathogenesis of type 2 diabetes mellitus (T2DM) remains insufficiently defined. This case–control study investigated associations between genetic variants in miRNA genes and susceptibility to T2DM in a Greek population. A total of 716 individuals with T2DM and 569 controls (HbA1c < 6.5% and fasting plasma glucose < 126 mg/dL) were included. Genomic DNA was extracted from whole blood and genotyped using the Illumina Infinium PsychArray platform. Polymorphisms in *MIR124a*, *MIR27a*, *MIR146a*, *MIR34a*, *MIRLET7A2*, *MIR128a*, *MIR196a2*, *MIR499a*, *MIR4513*, and *MIR149* were analyzed, with all SNPs within 20 kb upstream and downstream of each gene assessed. Allele frequencies were compared between cases and controls using PLINK. Significant associations with increased T2DM risk were observed for rs1531212 (OR = 1.375, *p* = 0.018) in *MIR23aHG* (containing *MIR27a*) and rs6120777 (OR = 1.27, *p* = 0.018) in *MYH7B*, upstream of *MIR499a*. Conversely, rs2425012 (OR = 0.794, *p* = 0.018) upstream of *MIR27a*, as well as rs883517 (OR = 0.728, *p* = 0.024) and rs2961920 (OR = 0.80, *p* = 0.041) upstream of *MIR146a*, appeared protective. Under the dominant model, two additional associations emerged: rs3746435 (OR = 1.239, *p* = 0.025) in *MYH7B* (upstream of *MIR499a*) and rs3746444 (OR = 1.235, *p* = 0.046) in *MIR499a*. In conclusion, this study identifies three novel genetic variants near *MIR27a* and *MIR499a* that may influence susceptibility to T2DM. These findings warrant validation in larger cohorts and functional studies to clarify their role in T2DM pathogenesis.

## 1. Introduction

Diabetes mellitus (DM) is a chronic disorder of glucose metabolism associated with impaired quality of life, severe complications, and a substantial burden on health care systems, and it has reached epidemic proportions worldwide. While type 1 diabetes mellitus (T1DM) typically manifests early in life, the incidence of type 2 diabetes mellitus (T2DM) increases progressively with age. According to the International Diabetes Federation, the global prevalence of DM is projected to reach 700 million by 2045. Diagnosis is based on plasma glucose or glycated hemoglobin (HbA1c) thresholds, established primarily through large-scale studies—mostly in T1DM cohorts—that evaluated the relationship between glycemic measures and diabetic complications, particularly microvascular organ damage [1]. Given the complex etiology of DM, genetic factors are recognized as important contributors to both hyperglycemia and its complications. Nevertheless, the precise genetic architecture of T2DM remains incompletely understood.

Recent advances in techniques such as genome-wide association studies (GWAS) have enabled the identification of numerous single-nucleotide polymorphisms (SNPs) and genomic loci associated with diabetes mellitus (DM) and related cardiometabolic traits [2]. A key limitation of genetic studies in this field is the lack of standardized criteria for defining cases and controls. In many GWAS, DM is classified using vague criteria, including self-reported diagnosis or the use of antidiabetic therapy, rather than uniform, clinically validated definitions.

MicroRNAs (miRNAs) are small non-coding RNAs, 19–24 nucleotides in length, that regulate gene expression by repressing translation or, to a lesser extent, by reducing mRNA stability through interactions with complementary sequences in the 3′ untranslated region (3′UTR) of target genes. The seed sequence, defined as nucleotides 2–8 of miRNA, plays a critical role in target recognition and regulation. Genetic variation within either the miRNA seed region or the complementary 3′UTR sequence of a target gene can impair binding specificity, leading to aberrant gene expression [3,4,5]. Both computational and experimental studies have demonstrated that variants in miRNA-binding sites may alter disease susceptibility by disrupting canonical recognition sites or creating novel binding sites [6,7,8]. Several studies have investigated the role of miRNAs in diabetes mellitus (DM) and obesity [9,10]. Although these studies have identified diverse variants affecting protein function and regulatory mechanisms, the application of miRNAs for target prediction or diagnosis remains limited by poor accuracy and high false-positive rates [11,12,13]. Furthermore, investigations of genomic variations in miRNA-binding sites are still relatively scarce. The aim of the present study was to investigate, for the first time in a Greek population, the association between genomic variations in miRNA seed sequences, their complementary sequences, or binding sites, and the risk of developing T2DM in elderly individuals (>65 years).

## 2. Results

### 2.1. Study Population Characteristics

A total of 1285 participants were included in the study: 716 individuals with T2DM (group A) and 569 controls (group B). Females predominated in both groups [group A: 52% vs. 48% males; group B: 62% vs. 38% males]. The mean age was 68.93 ± 9.53 years in group A and 73.46 ± 7.25 years in group B (*p* < 0.05). The mean duration of T2DM in group A was 14.39 ± 9.29 years. As expected, individuals with T2DM had significantly higher body weight (84.94 ± 16.85 vs. 79.00 ± 17.00 kg, *p* < 0.05), BMI (31.57 ± 5.43 vs. 29.82 ± 5.32 kg/m^2^, *p* < 0.05), FPG (153.15 ± 53.71 vs. 100.04 ± 13.52 mg/dL, *p* < 0.05), and HbA1c (7.29 ± 1.27% vs. 5.34 ± 0.56%, *p* < 0.05), compared with controls. Clinical and biochemical characteristics of the study population are summarized in Table 1.

### 2.2. Genetic Analysis

#### 2.2.1. Associations Between miRNA Gene Variants and T2D Risk—Primary Analysis

Genetic association analysis focused on SNPs located within 20 kb flanking regions of selected miRNA genes. Table 2 lists the miRNA genes examined, including chromosomal locations.

The primary analysis revealed significant differences in allele frequencies of several SNPs between individuals with T2DM (group A) and controls (group B). A total of five polymorphisms were identified as either conferring increased risk for T2DM or providing protection against the disease (Table 3).

Two variants were positively associated with T2DM. The first, rs1531212_C/T (p_perm_ = 0.02033, OR = 1.375, 95% CI = 1.049–1.802), is located within the precursor *MIR27a* gene on chromosome 19. This SNP represents a non-coding transcript variant in *MIR23aHG*, the host gene of the *MIR-23a/27a/24-2* cluster. The second variant, rs6120777_G/A (p_perm_ = 0.04054, OR = 1.27, 95% CI = 1.011–1.27), was identified as an intronic variant in the *MYH7B* gene on chromosome 20. *MYH7B* encodes myosin heavy chain 7B, a member of the myosin II motor-domain superfamily.

Conversely, three variants were found to exert a protective effect against T2DM. The first, rs2425012_G/A (p_perm_ = 0.01587, OR = 0.7945, 95% CI = 0.6602–0.956), is a synonymous variant in *MYH7B*. Although synonymous variants do not alter the protein sequence, increasing evidence suggests that they may still influence gene regulation and protein function. Two additional variants located near *MIR146a* also demonstrated protective associations. Rs883517_A/G (p_perm_ = 0.02675, OR = 0.7281, 95% CI = 0.5541–0.9567) is an intronic variant in *MIR3142HG*, the host gene of *MIR146a*. Rs2961920_A/C (p_perm_ = 0.04202, OR = 0.8057, 95% CI = 0.6548–0.9915) is also located within *MIR3142HG* and maps ~2 kb upstream of *MIR146a*. No additional variants reached statistical significance under the recessive model. Adjustment for sex revealed no further differences in allele frequencies.

#### 2.2.2. Dominant Genetic Model Analysis of SNP Associations with T2D—Secondary Analysis

In the secondary analysis, using a dominant genetic model, two additional variants were significantly associated with increased risk of T2DM (Table 4). The first, rs3746435_C/G (p_perm_ = 0.025, OR = 1.239, 95% CI = 1.001–1.535), is a missense variant in the *MYH7B* gene. The second, rs3746444_C/T (p_perm_ = 0.046, OR = 1.235, 95% CI = 0.9974–1.528), represents a non-coding transcript variant in *MIR499a* hosted in the *MYH7B* gene as shown in Figure 1. Adjustment for sex did not alter allele frequency distributions.

### 2.3. Target Gene and Pathway Analysis of Identified miRNA Variants

To elucidate the functional relevance of the identified polymorphisms, in silico analyses of *MIR27a*, *MIR499a*, and *MIR146a* target genes were performed using TargetScan, miRTarBase, and miRDB. Functional enrichment indicated convergence on pathways central to insulin signaling, mitochondrial energy metabolism, and inflammatory regulation, all crucial in T2DM pathogenesis.

*MIR27a*, containing the rs1531212 variant in the *MIR23aHG* host gene, regulates *PPARγ*, *IRS1*, *FOXO1*, and *PIK3R1*—core effectors of glucose uptake and adipocyte differentiation [14]. Perturbations in *MIR27a* expression can disrupt insulin receptor substrate phosphorylation and lipid handling, promoting insulin resistance. The rs1531212 variant may thus influence *MIR23aHG* transcription or pri-miRNA processing, modifying *MIR27a* levels and downstream metabolic signaling. *MIR499a*, encoded within *MYH7B* and encompassing risk variants rs6120777, rs3746435, and rs3746444, targets *PTEN*, *AMPKα2* (*PRKAA2*), *SIRT1*, and *PPARGC1A*—key regulators of mitochondrial biogenesis, oxidative phosphorylation, and β-cell survival [15,16,17]. By contrast, *MIR146a*, associated with protective variants rs883517 and rs2961920 in *MIR3142HG*, directly represses *IRAK1* and *TRAF6*, attenuating Toll-like receptor (TLR) and NF-κB signaling [18,19,20]. This negative feedback reduces inflammatory cytokine production and protects β-cells from cytokine-induced dedifferentiation and apoptosis [21].

Overall, these findings suggest that the miRNA variants identified here act as modulators of post-transcriptional control within three interlinked biological axes—insulin sensitivity, mitochondrial integrity, and inflammation—providing mechanistic plausibility for their respective risk and protective roles in T2DM.

## 3. Discussion

This study identified seven novel genetic variants located within or near *MIR27a*, *MIR499a*, and *MIR146a* loci as potential markers influencing susceptibility to T2DM in a Greek population. Four variants (rs1531212, rs6120777, rs3746435, and rs3746444) were associated with increased T2D risk, while three variants (rs2425012, rs883517, and rs2961920) appeared to exert a protective effect.

Two key variants within the *MYH7B* locus—rs6120777 and rs3746435—as well as rs3746444 in *MIR499a*, are of particular biological interest. *MYH7B* encodes a myosin heavy chain isoform implicated not only in hypertrophic cardiomyopathy but also in metabolic regulation [22]. The missense variant rs3746435 may alter myosin function, disrupting mitochondrial oxidative stress pathways known to contribute to insulin resistance and β-cell dysfunction. Similarly, rs3746444 in *MIR499a* has been linked to mitochondrial dysfunction and increased oxidative stress, reinforcing its role in T2D pathogenesis [23]. The variant rs1531212, located in the *MIR23aHG* host gene encompassing *MIR27a*, may modulate miRNA processing or expression, affecting pathways like insulin signaling and glucose metabolism.

Conversely, the synonymous rs2425012 variant in *MYH7B* may influence gene expression regulation via effects on mRNA stability or translation efficiency, though further functional validation is required. Protective variants rs883517 and rs2961920 lie within the *MIR3142HG* host gene upstream of *MIR146a*, a miRNA shown to regulate inflammatory signaling and β-cell homeostasis [24,25,26]. Recent studies suggest *MIR146a* variants can mitigate β-cell dedifferentiation under diabetic conditions, consistent with our findings of protective associations [27].

Overall, these results are consistent with prior evidence linking microRNA-related genes to T2DM. Variants in *MYH7B*, which hosts *MIR499a*, have been associated with both hypertrophic cardiomyopathy and oxidative stress–related metabolic pathways [28,29]. The rs3746444 variant, in particular, has been reported in previous studies as a T2DM susceptibility marker [30], reinforcing our findings. Likewise, *MIR146a* polymorphisms have shown heterogeneous effects across populations, being associated with both increased and reduced T2DM risk [31]. This dual pattern is consistent with our results, where *MIR146a*-related variants (rs883517 and rs2961920) appeared protective.

Growing evidence highlights the central role of microRNAs in the pathophysiology of diabetes and related cardiometabolic disorders. Integrative analyses have demonstrated that miRNAs, along with associated proteins and metabolites, constitute coordinated regulatory networks that mirror metabolic disturbances in prediabetes and diabetes [32]. Furthermore, polymorphisms within miRNA genes have been implicated in the development of both diabetes and atherosclerotic cardiovascular disease by influencing miRNA biogenesis and target specificity [33]. Consistent with these reports, our findings suggest that variants in *MIR27A*, *MIR499A*, and *MIR146A* may affect insulin signaling, mitochondrial homeostasis, and inflammatory pathways—mechanisms also highlighted in global miRNA profiling studies of diabetic tissues [34,35]. Collectively, these results reinforce the notion that miRNA genetic variation contributes to inter-individual differences in T2DM susceptibility and may represent a promising biomarker axis for early risk stratification and therapeutic targeting.

Importantly, the choice of a carefully characterized control group was crucial in identifying not only risk-associated but also protective variants. Including individuals without diabetes of comparable demographic and environmental background minimized confounding and allowed clearer attribution of genetic effects. This approach enabled the detection of protective polymorphisms such as rs883517 and rs2961920 in *MIR3142HG*, which were more frequent among controls.

This study has several strengths. Its targeted focus on microRNA-related regions highlights key regulatory loci involved in T2DM, while the identification of both risk and protective alleles provides a more comprehensive view of genetic susceptibility. Furthermore, the well-characterized control group enhances the reliability of the findings. However, some limitations should be acknowledged. First, the sample size was relatively modest, which may limit statistical power and generalizability, underscoring the need for replication in larger, multi-ethnic cohorts. Second, the cross-sectional, observational design precludes causal inference. Third, some individuals in the control group fell within the prediabetic range; therefore, the possibility that they may develop T2DM later in life cannot be excluded. However, given the relatively advanced mean age of the control population, this limitation is unlikely to have substantially influenced the results. Finally, no functional assays were performed, leaving the biological mechanisms through which these variants act unresolved.

## 4. Materials and Methods

### 4.1. Ethical Considerations

The study protocol was approved by the Ethics Committee of the Aristotle University of Thessaloniki (approval number A6452/date 2 March 2011). Written informed consent was obtained from all participants prior to enrollment. All procedures were conducted in accordance with the ethical standards of the 1964 Declaration of Helsinki and its later amendments.

### 4.2. Study Population

Participants aged > 65 years with a confirmed diagnosis of T2DM (group A, *n* = 716) were recruited from two major academic diabetes centers in Greece: AHEPA University Hospital of Thessaloniki and the General University Hospital of Alexandroupolis. Diagnosis of T2DM was established according to the American Diabetes Association criteria [1]. Exclusion criteria for all participants covered severe hepatic diseases and current treatment with medications known to increase blood glucose levels, such as corticosteroids and antipsychotics, to limit confounding effects on glycemic measurements. Patients with a history of cancer were not specifically excluded, as this was not the primary focus of the present study; however, no participants had active malignancy at the time of enrollment.

Healthy individuals were recruited as controls (group B, *n* = 569). Inclusion criteria were (a) age >65 years, (b) HbA1c <6.5% (48 mmol/mol), (c) fasting plasma glucose (FPG) <126 mg/dL, (d) no history of T2DM, (e) no past or current use of glucose-altering agents, and (f) no current use of medications known to affect blood glucose levels, as described above. Metabolic syndrome was not an official exclusion criterion for controls. Nevertheless, the mean HbA1c and fasting glucose values of the control group indicate that these individuals were metabolically healthy despite their advanced age, supporting the robustness of the control selection. All participants in both groups were of Greek ethnicity and origin.

### 4.3. Clinical and Biochemical Measurements

Study participants completed a questionnaire to provide demographic information. Clinical data, including family history of T2DM, age at diagnosis, ancestry, and current and past medication use, were collected from self-reported information and medical records. Anthropometric measurements were obtained with participants wearing light clothing and no shoes. Height was measured using a wall-mounted stadiometer, and body weight was recorded with a calibrated scale. Body mass index (BMI) was calculated as weight (kg) divided by height (m^2^). Waist circumference (cm) was measured at the midpoint between the 12th rib and the iliac crest.

After a 12 h overnight fast, morning blood samples were collected by antecubital venipuncture and stored at −20 °C until analysis. Plasma glucose levels were measured using the Cobas INTEGRA clinical chemistry analyzer (Roche Diagnostics, Mannheim, Germany). The reference range for glucose was 70–110 mg/dL, with inter- and intra-assay coefficients of variation of 0.99% and 3.5%, respectively. Levels of HbA1c were determined using the ADAMS HA-8160 high-performance liquid chromatography (HPLC) system (A. Menarini Diagnostics, Florence, Italy).

### 4.4. DNA Extraction and Genotyping

Genomic DNA was extracted from peripheral blood (QIAamp DNA blood kit; QIAGEN, Hilden, Germany), and samples were genotyped on Illumina Infinium PsychArray [603132 Single-Nucleotide Variants (SNPs), out of which 559,921 are submitted to dbSNP as rs]. The initial SNP call rate threshold used was 0.95. As part of quality control for the individuals, all individuals with a call rate less than 0.98, an absolute value of inbreeding coefficients over 0.2, and a gender phenotype that deviated from their genotypic sex were removed. Similarly, for SNP quality control, all SNPs that failed the call rate threshold of 0.98, the call rate difference threshold of 0.02 between individuals with T2DM and controls, and a Hardy–Weinberg Equilibrium *p*-value of 10^−6^ for controls and 10^−10^ for individuals with T2DM were left out. Monomorphic markers were kept for the filtering stage but removed later, during the subsequent parts of the analysis. Genetic relatedness (identity-by-descent) and principal components (eigenvectors) were calculated after the completion of quality control that consisted of the removal of any outliers and one individual out of every pair that demonstrated a genetic relatedness ratio over 0.1875. Genotypic data were processed and analyzed using PLINK v1.9 [36,37] and EIGENSOFT v7.2 [2]. Variants located within or ±20 kb of the selected microRNA loci (*MIR124A*, *MIR27A*, *MIR146A*, *MIR34A*, *MIRLET7A2*, *MIR128A*, *MIR196A2*, *MIR499A*, *MIR4513*, and *MIR149*) were extracted to capture both intragenic and cis-regulatory regions potentially influencing miRNA transcription or maturation.

Quality control (QC) procedures were applied to ensure high-confidence genotype calls. Variants with a minor allele frequency (MAF) < 0.05, genotyping call rate < 95%, or deviation from Hardy–Weinberg equilibrium (*p* < 0.001) were excluded. Samples with genotype missingness > 5% or extreme heterozygosity were also removed. Population structure was evaluated using principal component analysis (PCA) implemented in EIGENSOFT to confirm genetic homogeneity and exclude population outliers. The top principal components were examined and, where necessary, included as covariates to correct for subtle stratification. For association testing, logistic regression models were implemented in PLINK under an additive genetic model, with age, sex, and BMI included as covariates. Complementary allelic, dominant, and recessive models were also explored to assess model robustness. Odds ratios (ORs) and 95% confidence intervals (CIs) were reported to estimate effect size and direction of association.

All genomic positions were referenced to the GRCh37/hg19 human genome build to ensure compatibility across tools and annotation databases.

### 4.5. Statistical Analysis

All statistical analyses were performed using PLINK v1.9 and SPSS software (IBM SPSS Statistics for Windows, Version 21.0; IBM Corp., Foster City, CA, USA). Differences between patient and control groups were assessed using Student's *t*-test or Pearson’s chi-square test. Data are presented as mean ± standard deviation. A two-tailed *p*-value <0.05 was considered statistically significant.

Genetic data analyses were conducted in PLINK. Logistic regression was performed with T2DM status as the outcome variable and sex as a covariate. A second regression analysis was conducted with the same settings but additionally adjusting for BMI and waist circumference. Because markers within specific genomic regions are often in high linkage disequilibrium (LD), significance levels were assessed using permutation testing to account for multiple comparisons. Bonferroni correction was not applied, as it can result in inflated false-negative rates by overly reducing the nominal type I error rate. Given the correlation among SNPs in the same gene region, permutation testing provides a more appropriate approach. This procedure involves random shuffling of phenotypes and estimation of empirical *p*-values (p_perm_). The adaptive permutation method implemented in PLINK was used, which terminates permutations for SNPs unlikely to reach significance. This method has been reported as one of the most effective strategies for multiple testing correction in gene-based analyses [36,37]. Monomorphic SNPs were excluded from all analyses. Figure 2 presents the study workflow.

## 5. Conclusions

In summary, this study identified seven novel genetic variants in microRNA-related regions associated with T2DM susceptibility. Four variants near *MIR27a* and *MIR499a* (rs1531212, rs6120777, rs3746435, and rs3746444) were associated with increased risk, while three variants near *MIR27a* and *MIR146a* (rs2425012, rs883517, and rs2961920) appeared protective. Variants in *MYH7B* and *MIR3142HG* emerged as central contributors, potentially modulating oxidative stress regulation and β-cell function. These findings broaden the current understanding of microRNA-related genetic variation in T2DM and highlight novel pathways involved in disease pathogenesis.

Future research should aim to replicate these associations in larger, multi-ethnic cohorts and perform functional studies to elucidate the underlying biological mechanisms. In addition, exploration of gene–environment interactions and evaluation of their potential for clinical application in genetic risk stratification and therapeutic targeting are warranted.

## Figures and Tables

**Figure 1 ijms-26-10447-f001:**
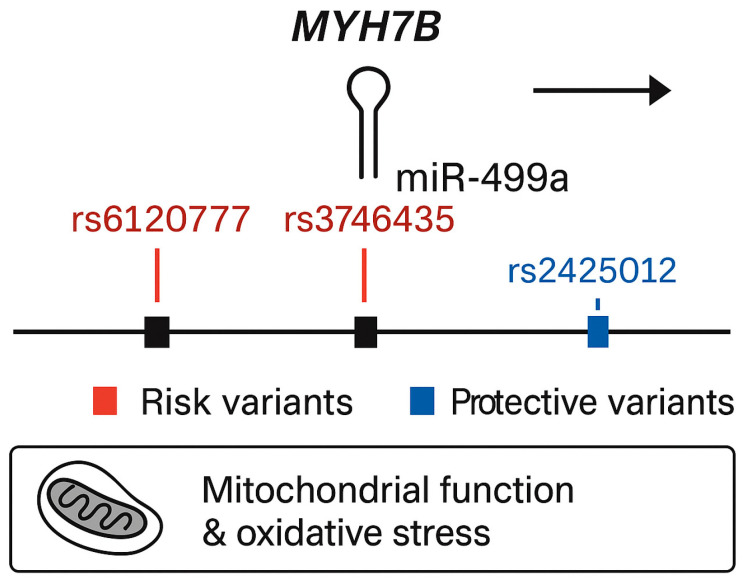
Genomic structure of the MYH7B locus showing the embedded miR-499a and associated variants identified in the present study.

**Figure 2 ijms-26-10447-f002:**
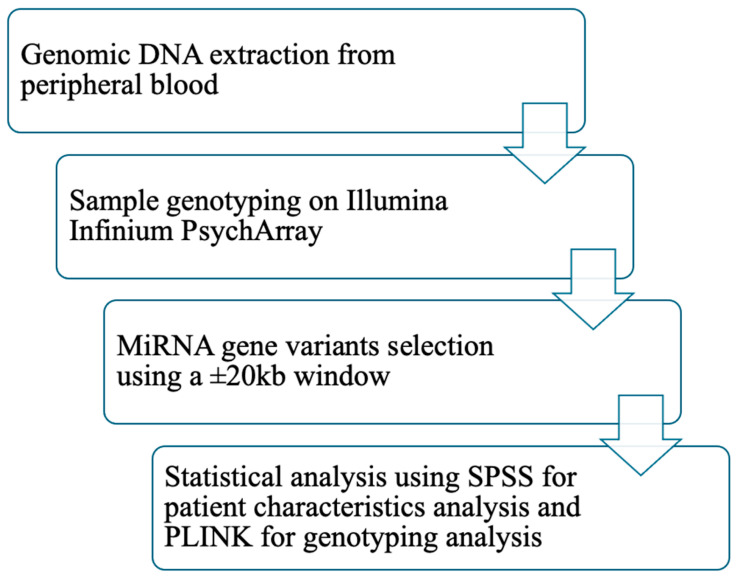
Workflow of the study. Abbreviations: DNA: deoxyribonucleic acid; miRNA: micro ribonucleic acid; kb: kilobase; SPSS: statistical package for the social sciences; PLINK: providing linkage and integrated knowledge.

**Table 1 ijms-26-10447-t001:** Patient characteristics and between-group comparisons.

Patient Characteristics	Group A–Diabetes (N = 716)	Group B–Control (N = 569)	Group A vs. Group B (*p*-Value)
Age (years)	68.93 (±9.53)	73.46 (±7.25)	0.001
Gender (M/F)	48%:52%	38%:62%	0.001
Body weight (kg)	84.94 (±16.85)	79.00 (±17.00)	0.001
BMI (kg/m^2^)	31.57 (±5.43)	29.82 (±5.32)	0.001
Waist circumference (cm)	104.62 (±15.031)	102.02 (±11.88)	0.001
Hb1Ac (%)	7.29 (±1.27)	5.34 (±0.56)	0.001
Fasting glucose (mg/dL)	153.15 (±53.71)	100.04 (±13.51)	0.001
Diabetes duration (years)	14.39 (±9.29)	-	-

Data are presented as means ± SD. Abbreviations: BMI: body mass index; M: males; F: females; Hb1Ac: hemoglobin 1Ac. Standard values: BMI: underweight (below 18.5 kg/m^2^), healthy weight (18.5 to 24.9 kg/m^2^), overweight (25.0 to 29.9), and obesity (30.0 kg/m^2^ or higher); Hb1Ac: normal: below 5.7%, prediabetes: 5.7% to 6.4%, and diabetes: 6.5% or higher.

**Table 2 ijms-26-10447-t002:** The miRNA genes studied.

Gene	Chr	Start (hg19)	End (hg19)	GeneCard Gene Name
*MIR124a*	8	9757574	9762876	*MIR124-1*
*MIR146a*	5	159895275	159914433	*MIR146a*
*MIR27a*	19	13947254	13947331	*MIR27a*
*MIR34a*	1	9211727	9211836	*MIR34a*
*MIRLET7A2*	11	122017229	122017301	*MIRLET7A2*
*MIR128a*	2	136422967	136423048	*MIR128-1*
*MIR196a2*	12	54385522	54385631	*MIR196a2*
*MIR499*	20	33578179	33578300	*MIR499A*
*MIR4513*	15	75081013	75081098	*MIR4513*
*MIR149*	2	241395418	241395506	*MIR149*

**Table 3 ijms-26-10447-t003:** Primary analysis results.

Gene	Chromosome	SNP	Position	Recessive Allele	Dominant Allele	p_perm_	MAF	OR	Lower 95% CI	Upper 95% CI
*MIR27a*	19	rs1531212	13951830	T	C	0.02033	0.147	1.375	1.049	1.802
*MYH7B*	20	rs6120777	33560172	A	G	0.04054	0.217	1.27	1.011	1.596
*MYH7B*	20	rs2425012	33581955	A	G	0.01587	0.441	0.7945	0.6602	0.956
*upstream of MIR146a*	5	rs883517	159904729	G	A	0.02675	0.123	0.7281	0.5541	0.9567
*upstream of MIR146a*	5	rs2961920	159911506	C	A	0.04202	0.260	0.8057	0.6548	0.9915

Abbreviations: MAF: major allele frequency; SNP: single-nucleotide polymorphism; OR: odds ratio; CI: confidence interval. In red: OR > 1: The odds of the outcome are higher in the study group compared to the control group. A higher OR indicates a stronger association.

**Table 4 ijms-26-10447-t004:** Secondary analysis results.

Gene	Chromosome	SNP	Position	Recessive Allele	Dominant Allele	p_perm_	MAF	OR	Lower 95% CI	Upper 95% CI
*MYH7B*	20	rs3746435	33587198	C	G	0.025	0.2589	1.239	1.001	1.535
*MIR499a*	20	rs3746444	33578251	C	T	0.046	0.26	1.235	0.9974	1.528

Abbreviations: MAF: major allele frequency; SNP: single-nucleotide polymorphism; OR: odds ratio; CI: confidence interval. In red: OR > 1: The odds of the outcome are higher in the study group compared to the control group. A higher OR indicates a stronger association.

## Data Availability

The original contributions presented in this study are included in the article. Further inquiries can be directed to the corresponding author.

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
