# Peer review of "Strong Association Between MiRNA Gene Variants and Type 2 Diabetes Mellitus in a Caucasian Population"

_ijms, 2025, doi:10.3390/ijms262110447_

Round 1

Reviewer 1 Report

Comments and Suggestions for Authors

the manuscript by Manthou et al., described association of microRNAs gene polymorphism in MIR124a, MIR27a, MIR146a, MIR34a, MIRLET7A2, MIR128a, MIR196a2, MIR499a, MIR4513, and MIR149 were analyzed. they conclude that MIR27a and MIR499a may be associated with T2D development. 

-in abstract the authors should include more information on the genotyping methodology 

I have the following suggestions that can be useful to improve this manuscript.

-the authors should provide more details and citations of the genotyping methodology and statistical analysis. 

-the authors should explain why only the patients with liver diseases and people taking medicine that increase blood sugar excluded from the study. What about patients with cancers and diseases of metabolic syndrome? are they included ?.

-the authors should put the ethical approval information at the start of the Materials and methods paragraph in my opinion.

-if possible please include more patients data in table 1

-discussion.

-the authors should explain how the rs1531212, rs6120777, rs3746435, and rs3746444 are associated with increased susceptibility to T2D, for example induction of insulin resistance 

-the authors should discuss how the rs883517 and rs2961920 protect against T2D.

-the following paper should be discussed 

- MicroRNA, Proteins, and Metabolites as Novel Biomarkers for Prediabetes, Diabetes, and Related Complications, 10.3389/fendo.2018.00180

-Potential Impact of MicroRNA Gene Polymorphisms in the Pathogenesis of Diabetes and Atherosclerotic Cardiovascular Disease, doi: 10.3390/jpm9040051

- The Profiling and Role of miRNAs in Diabetes Mellitus, 10.33696/diabetes.1.003

Author Response

Response to Reviewer 1 Comments

1. Summary

Thank you very much for taking the time to review this manuscript. Please find the detailed responses below and the corresponding revisions/corrections highlighted/in track changes in the re-submitted file.

2. Point-by-point response to Comments and Suggestions for Authors

Comments 1: in abstract the authors should include more information on the genotyping methodology

Response 1: Thank you for pointing this out. In our abstract we include the following text regarding the genotyping methodology “Genomic DNA was extracted from whole blood and genotyped using the Illumina Infinium PsychArray platform. Polymorphisms in MIR124a, MIR27a, MIR146a, MIR34a, MIRLET7A2, MIR128a, MIR196a2, MIR499a, MIR4513, and MIR149 were analyzed, with all SNPs within 20 kb upstream and downstream of each gene assessed. Allele frequencies were compared between cases and controls using PLINK.” and then we proceed in further details in the materials and methods section.

Comments 2: the authors should provide more details and citations of the genotyping methodology and statistical analysis

Response 2: We fully agree. We have, accordingly, revised and modified this section of the paper to emphasize this point. You will find the revised text in section 4.4 DNA extraction and Genotyping “Genotypic data were processed and analyzed using PLINK v1.9 [37,38] and EIGENSOFT v7.2 [39]. Variants located within or ±20 kb of the selected microRNA loci (MIR124A, MIR27A, MIR146A, MIR34A, MIRLET7A2, MIR128A, MIR196A2, MIR499A, MIR4513, and MIR149) were extracted to capture both intragenic and cis-regulatory regions potentially influencing miRNA transcription or maturation.

Quality control (QC) procedures were applied to ensure high-confidence genotype calls. Variants with a minor allele frequency (MAF) < 0.05, genotyping call rate < 95%, or deviation from Hardy–Weinberg equilibrium (p < 0.001) were excluded. Samples with genotype missingness > 5% or extreme heterozygosity were also removed. Population structure was evaluated using principal component analysis (PCA) implemented in EIGENSOFT to confirm genetic homogeneity and exclude population outliers. The top principal components were examined and, where necessary, included as covariates to correct for subtle stratification. For association testing, logistic regression models were implemented in PLINK under an additive genetic model, with age, sex, and BMI included as covariates. Complementary allelic, dominant, and recessive models were also explored to assess model robustness. Odds ratios (ORs) and 95% confidence intervals (CIs) were reported to estimate effect size and direction of association. 

All genomic positions were referenced to the GRCh37/hg19 human genome build to ensure compatibility across tools and annotation databases.”

Comments 3: the authors should explain why only the patients with liver diseases and people taking medicine that increase blood sugar excluded from the study. What about patients with cancers and diseases of metabolic syndrome? are they included?

Response 3: Thank you for this thoughtful comment. In the Materials and Methods section, we have clarified the exclusion criteria as follows:

“Participants with severe hepatic disease or receiving medications known to increase blood glucose (e.g., corticosteroids, antipsychotics) were excluded to avoid confounding glycemic measurements.”

We did not exclude individuals with a history of cancer, as this was not the focus of the present study; however, none of the participants had active cancer at the time of enrollment. Regarding metabolic syndrome, although it was not an official exclusion criterion for controls, the mean HbA1c and fasting glucose values of the control group indicate that these individuals were metabolically healthy despite their advanced age. This supports the robustness of the control selection and the reliability of between-group comparisons. The Methods section has been revised accordingly to clearly reflect these inclusion and exclusion criteria.

Comments 4: the authors should put the ethical approval information at the start of the Materials and methods paragraph in my opinion.

Response 4: We fully agree with your comment regarding the most appropriate structure of the paper and we changed it in our revised version so now you will find section 4.1 Ethical considerations in the beginning of section 4. Materials and Methods.

Comment 5: if possible please include more patients data in table 1

Response 5: We agree that more detailed phenotypic data would be valuable; however, all available and reliable clinical and biochemical variables collected through the standardized questionnaire are presented in Table 1 to maintain data completeness and comparability between groups.

Comment 6 & 7:

●      the authors should explain how the rs1531212, rs6120777, rs3746435, and rs3746444 are associated with increased susceptibility to T2D, for example induction of insulin resistance 

●      the authors should discuss how the rs883517 and rs2961920 protect against T2D

Response 6 & 7: 

Thank you for this insightful comment. Based on your comment we added information addressing the associations of these SNPs with T2DM found in the current literature as part of a target gene analysis. You can find this in the results section paragraph 2.3 Target Gene and Pathway Analysis of Identified miRNA Variants

“To elucidate the functional relevance of the identified polymorphisms, in silico analyses of MIR27aMIR499a, and MIR146a target genes were performed using TargetScan, miRTarBase, and miRDB. Functional enrichment indicated convergence on pathways central to insulin signaling, mitochondrial energy metabolism, and inflammatory regulation, all crucial in T2DM pathogenesis.

MIR27a, containing the rs1531212 variant in the MIR23aHG host gene, regulates PPARγIRS1FOXO1, and PIK3R1—core effectors of glucose uptake and adipocyte differentiation [13,14]. Perturbations in MIR27a expression can disrupt insulin receptor substrate phosphorylation and lipid handling, promoting insulin resistance. The rs1531212 variant may thus influence MIR23aHG transcription or pri-miRNA processing, modifying MIR27a levels and downstream metabolic signaling. MIR499a, encoded within MYH7B and encompassing risk variants rs6120777, rs3746435, and rs3746444, targets PTENAMPKα2 (PRKAA2), SIRT1, and PPARGC1A—key regulators of mitochondrial biogenesis, oxidative phosphorylation, and β-cell survival [15-17]. By contrast, MIR146a, associated with protective variants rs883517 and rs2961920 in MIR3142HG, directly represses IRAK1 and TRAF6, attenuating Toll-like receptor (TLR) and NF-κB signaling [18-20]. This negative feedback reduces inflammatory cytokine production and protects β-cells from cytokine-induced dedifferentiation and apoptosis [21].

Overall, these findings suggest that the miRNA variants identified here act as modulators of post-transcriptional control within three interlinked biological axes—insulin sensitivity, mitochondrial integrity, and inflammation—providing mechanistic plausibility for their respective risk and protective roles in T2DM.”

Comment 8: the following paper should be discussed 

●      MicroRNA, Proteins, and Metabolites as Novel Biomarkers for Prediabetes, Diabetes, and Related Complications, 10.3389/fendo.2018.00180

●      Potential Impact of MicroRNA Gene Polymorphisms in the Pathogenesis of Diabetes and Atherosclerotic Cardiovascular Disease, doi: 10.3390/jpm9040051

●      The Profiling and Role of miRNAs in Diabetes Mellitus, 10.33696/diabetes.1.003

Response 8: We are deeply thankful for taking the time to study our paper as well as the bibliography thoroughly. Your paper suggestions are very helpful to fully accommodate our cause. These are all addressed in the discussion section in a new paragraph which we quote  “Growing evidence highlights the central role of microRNAs in the pathophysiology of diabetes and related cardiometabolic disorders. Integrative analyses have demonstrated that miRNAs, along with associated proteins and metabolites, constitute coordi-nated regulatory networks that mirror metabolic disturbances in prediabetes and diabetes [34]. Furthermore, polymorphisms within miRNA genes have been implicated in the development of both diabetes and atherosclerotic cardiovascular disease by influencing miRNA biogenesis and target specificity [35]. Consistent with these reports, our findings suggest that variants in MIR27A, MIR499A, and MIR146A may affect insulin signaling, mitochondrial homeostasis, and inflammatory pathways—mechanisms also highlighted in global miRNA profiling studies of diabetic tissues [36]. Collectively, these results reinforce the notion that miRNA genetic variation contributes to inter-individual differences in T2DM susceptibility and may represent a promising biomarker axis for early risk stratification and therapeutic targeting.”

Please ignore the file uploaded it was a pre-edited manuscript of our response. 

Thank you in advance! 

Reviewer 2 Report

Comments and Suggestions for Authors

The manuscript by Manthou et al. sets out to investigate the association between genetic variants in miRNA genes and type 2 diabetes (T2D) susceptibility in a cohort of Greek origin consisting of 716 individuals with T2D and 569 controls.

First the authors report differences in allele frequencies between individuals with T2D and controls. (Table not shown?)

Afterwards the group around Manthou et al. identifies 5 Single Nucleotide Polymorphisms (SNPs) as either susceptible or protective for the risk of developing T2D using a recessive genetic model.  Here 2 out of the 5 variants were detected to be risk variants whereas the other 3 detected variants exerted a protective effect. In addition, the authors connect 2 additional variants with an increased risk of developing T2D when using a dominant genetic model.

I congratulate the authors on a very interesting study. Nevertheless, I would like the authors to address the following concerns to enhance the clarity of the data.

1.I would like to suggest the authors to replace the term patients with individuals with T2D in the abstract l18 and throughout the manuscript

2. Please give standard values in table 1 where they are applicable like BMI, Hba1c, etc.

3. ll 176/177 the primary analysis revealed significant differences in allele frequencies of several SNPs between individuals and controls (Table 2) – I am not able to detect the differences in allele frequencies in table 2 to me it just lists the different miRNAs studied

4. I would like to suggest the authors to actually rename the headings of the paragraphs with the actual result demonstrated and not just Primary analysis, Secondary analysis

5. Table 3 and 4 explain what it means when numbers are marked in red in the figure legenand put also the Chromosome as a column

6. I would like the authors to add a target gene analysis for the different miRNAs where variants were either identified as risk or protective variant. Are those target genes T2D risk genes and in which pathways are they involved

7.l250 the information MYH7B hosts miR499a should be given earlier to the reader or a Figure should be created where the locus and variants are displayed to make it easier to understand for the reader. 

Author Response

Response to Reviewer 2 Comments

1. Summary

Thank you very much for taking the time to review this manuscript. Please find the detailed responses below and the corresponding revisions/corrections highlighted/in track changes in the re-submitted file.

2. Point-by-point response to Comments and Suggestions for Authors

Comments 1: I would like to suggest the authors to replace the term patients with individuals with T2D in the abstract l18 and throughout the manuscript

Response 1: Thank you for pointing this out. All mentions of "patients" were revised to "individuals with type 2 diabetes mellitus (T2DM)" to ensure precision and respectfulness. These changes were implemented consistently throughout the manuscript, tables, and figure legends to maintain uniform terminology. You can find them highlighted in the revised form. 

Comments 2: Please give standard values in table 1 where they are applicable like BMI, Hba1c, etc.

Response 2: We fully agree. We have, accordingly, revised and modified the explanations under Table 1 to include standard values. We therefore included the following additions “Standard values: BMI: underweight (below 18.5 kg/m2), healthy weight (18.5 to 24.9 kg/m2), overweight (25.0 to 29.9 kg/m2), and obesity (30.0 kg/m2 or higher); Hb1Ac: Normal: below 5.7%, Prediabetes: 5.7% to 6.4%, and Diabetes: 6.5% or higher”

Comments 3: 176/177 the primary analysis revealed significant differences in allele frequencies of several SNPs between individuals and controls (Table 2) – I am not able to detect the differences in allele frequencies in table 2 to me it just lists the different miRNAs studied

Response 3: We sincerely thank you for pointing this out. It was indeed a misinterpretation. The explanation for table 2 was corrected and highlighted in the text “Table 2 lists the miRNA genes examined, including chromosomal locations.”

Comments 4: I would like to suggest the authors to actually rename the headings of the paragraphs with the actual result demonstrated and not just Primary analysis, Secondary analysis

Response 4: We fully agree with your comment regarding the most appropriate titles for these sections thus we changed it in our revised version so now you find that Sections 2.2.1 and 2.2.2 are now titled: "2.2.1 Associations between miRNA gene variants and T2D risk – Primary analysis" and "2.2.2 Dominant genetic model analysis of SNP associations with T2D – Secondary analysis"

Comment 5: Table 3 and 4 explain what it means when numbers are marked in red in the figure legend and put also the Chromosome as a column

Response 5: We thank the reviewer for the helpful suggestion. In table 3 and 4 red in the figure is explained in the table label “In red: OR > 1: The odds of the outcome are higher in the study group compared to the control group. A higher OR indicates a stronger association.” and an extra column with the chromosome of its gene is included.

Comment 6: I would like the authors to add a target gene analysis for the different miRNAs where variants were either identified as risk or protective variant. Are those target genes T2D risk genes and in which pathways are they involved.

Response 6: 

Thank you for this insightful comment. We appreciate that both reviewers highlighted this point, and have thus expanded the relevant sections to better integrate miRNA functional networks into our interpretation of genetic associations. Specifically: 

“To elucidate the functional relevance of the identified polymorphisms, in silico analyses of MIR27aMIR499a, and MIR146a target genes were performed using TargetScan, miRTarBase, and miRDB. Functional enrichment indicated convergence on pathways central to insulin signaling, mitochondrial energy metabolism, and inflammatory regulation, all crucial in T2DM pathogenesis.

MIR27a, containing the rs1531212 variant in the MIR23aHG host gene, regulates PPARγIRS1FOXO1, and PIK3R1—core effectors of glucose uptake and adipocyte differentiation [13,14]. Perturbations in MIR27a expression can disrupt insulin receptor substrate phosphorylation and lipid handling, promoting insulin resistance. The rs1531212 variant may thus influence MIR23aHG transcription or pri-miRNA processing, modifying MIR27a levels and downstream metabolic signaling. MIR499a, encoded within MYH7B and encompassing risk variants rs6120777, rs3746435, and rs3746444, targets PTENAMPKα2 (PRKAA2), SIRT1, and PPARGC1A—key regulators of mitochondrial biogenesis, oxidative phosphorylation, and β-cell survival [15-17]. By contrast, MIR146a, associated with protective variants rs883517 and rs2961920 in MIR3142HG, directly represses IRAK1 and TRAF6, attenuating Toll-like receptor (TLR) and NF-κB signaling [18-20]. This negative feedback reduces inflammatory cytokine production and protects β-cells from cytokine-induced dedifferentiation and apoptosis [21]. 

Overall, these findings suggest that the miRNA variants identified here act as modulators of post-transcriptional control within three interlinked biological axes—insulin sensitivity, mitochondrial integrity, and inflammation—providing mechanistic plausibility for their respective risk and protective roles in T2DM.”

Comment 7: l250 the information MYH7B hosts miR499a should be given earlier to the reader or a Figure should be created where the locus and variants are displayed to make it easier to understand for the reader.

Response 7: We are deeply thankful for taking the time to study our paper thoroughly in order to point out any possible missings that can lead to potentials misconceptions. Thus we revised the part of the paper where the MYH7B gene is analyzed (section 2.2.2 Dominant Genetic Model Analysis of SNP associations with T2D – Secondary Analysis) adding an appropriate explanation and a figure to accompany it: “The second, rs3746444_C/T (pperm = 0.046, OR = 1.235, 95% CI = 0.9974–1.528), represents a non-coding transcript variant in MIR499a hosted in the MYH7B gene as shown in Figure 1.” This figure aids in visualizing the genomic organization of the MYH7B locus and clarifies the spatial relationship of the identified variants, thereby improving interpretability for readers unfamiliar with miRNA host-gene architecture.

Round 2

Reviewer 1 Report

Comments and Suggestions for Authors

Yhank you very much. the authors addressed my questions 

Reviewer 2 Report

Comments and Suggestions for Authors

No further comments